# Conceptualizing the COVID-19 Pandemic: Perspectives of Pregnant and Lactating Women, Male Community Members, and Health Workers in Kenya

**DOI:** 10.3390/ijerph191710784

**Published:** 2022-08-30

**Authors:** Alicia M. Paul, Clarice Lee, Berhaun Fesshaye, Rachel Gur-Arie, Eleonor Zavala, Prachi Singh, Ruth A. Karron, Rupali J. Limaye

**Affiliations:** 1International Vaccine Access Center, Bloomberg School of Public Health, Johns Hopkins University, Baltimore, MD 21205, USA; 2Department of Health, Behavior, and Society, Bloomberg School of Public Health, Johns Hopkins University, Baltimore, MD 21205, USA; 3Department of International Health, Bloomberg School of Public Health, Johns Hopkins University, Baltimore, MD 21205, USA; 4Berman Institute of Bioethics, Bloomberg School of Public Health, Johns Hopkins University, Baltimore, MD 21205, USA; 5Center for Immunization Research, Department of International Health, Bloomberg School of Public Health, Johns Hopkins University, Baltimore, MD 21205, USA; 6Department of Epidemiology, Bloomberg School of Public Health, Johns Hopkins University, Baltimore, MD 21205, USA

**Keywords:** COVID-19, health behavior, qualitative, maternal health, pregnancy, Kenya

## Abstract

Pregnant women are at greater risk of adverse outcomes from SARS-CoV-2 infection. There are several factors which can influence the ways in which pregnant women perceive COVID-19 disease and behaviorally respond to the pandemic. This study seeks to understand how three key audiences—pregnant and lactating women (PLW), male community members, and health workers—in Kenya conceptualize COVID-19 to better understand determinants of COVID-19 related behaviors. This study used qualitative methods to conduct 84 in-depth interviews in three counties in Kenya. Data were analyzed using a grounded theory approach. Emerging themes were organized based on common behavioral constructs thought to influence COVID-19 related behaviors and included myths, risk perception, economic implications, stigma, and self-efficacy. Results suggest that risk perception and behavioral attitudes substantially influence the experiences of PLW, male community members, and health workers in Kenya during the COVID-19 pandemic. Public health prevention and communication responses targeting these groups should address potential barriers to preventive health behaviors, such as the spread of misinformation, financial constraints, and fear of social ostracization.

## 1. Introduction

SARS-CoV-2 has impacted every corner of the world. In Kenya, the first case of SARS-CoV-2 infection was reported on 13 March 2020 [1]. One month later, Kenya closed its schools, restricted travel in and out of its capital city, Nairobi [1], and implemented a nationwide dusk-to-dawn curfew [2]. Since then, Kenya has seen epidemic peaks in July 2020, November 2020, and March 2021. By August 2021, Kenya was reporting approximately 3700 confirmed cases per 1 million people and 72 deaths due to COVID-19 per 1 million people [3].

In a study of nearly 1600 pregnant and postpartum women in Kenya, Otieno et al. (2021) found that within a nine-month period, COVID-19-like illness developed in up to 33% of participants [4]. Several studies point to particular populations at greater risk of adverse outcomes, including pregnant women. Pregnant women are at high risk of severe complications from SARS-CoV-2 infection, with increased need for hospitalization, intensive care, or ventilatory support compared to non-pregnant populations [5,6,7,8]. Pregnancy outcomes have also been impacted by SARS-CoV-2 infection. Risk of preterm birth, stillbirth, and other pregnancy complications are heightened by COVID-19, as well as preeclampsia, which can result in serious illness or death for the mother and baby [9,10].

The ways in which people conceptualize COVID-19 disease vary on an individual level. Factors such as socioeconomic status, social networks, individual experiences, and knowledge, attitudes, or beliefs about COVID-19 might shape a person’s view of the pandemic in terms of their risk-perception, optimism, trust in public health authorities, and economic stability, among other variables [11,12,13]. It is well established in behavioral health literature that psychological, social, and environmental factors affect behavioral intentions and behaviors themselves [14,15]. Many theories have been derived to explain the ways in which individuals determine how to behave, or their behavior change pathways, such as Social Cognitive Theory, the Theory of Planned Behavior, and the Health Belief Model. Across these theoretical perspectives are common constructs which are considered to influence behaviors across health topics.

Several constructs are particularly relevant across all health behaviors. For example, risk perception, or one’s beliefs about potential harm, has been used across several behavioral theories as a key influence of individual behavior [16]. In the Health Belief Model, risk perception is divided into two constructs: perceived susceptibility and perceived severity [17]. According to this model, the combination of perceived susceptibility and severity, among other factors, contribute to a person’s likelihood of engaging in a specified health behavior through the beliefs that there is at least some possibility of contracting an illness and that the illness can cause undesirable clinical or social consequences [17]. Another common construct for predicting health behaviors is self-efficacy, or “perceived behavioral control”, as referenced by the Theory of Planned Behavior. According to this theory, self-efficacy, or a person’s perception of their ability to perform a specified behavior, can influence a person’s health behaviors by minimizing their motivation to engage in that behavior out of fear that their chances of success are minimal [18]. Factors which might influence one’s self-efficacy include perceptions of physical ability, access to information, resources, or time, and economic capacity, among others. Moreover, the Theory of Planned Behavior posits that attitudes, or the degree to which a person favors a behavior of interest, can be highly predictive of behavioral intentions when also accounting for social norms and behavioral control [18]. More specifically, it is thought that attitudes are a function of one’s beliefs toward a target behavior, such that the degree to which a person believes that a behavior of interest will lead to a particular outcome will influence whether they have a favorable or unfavorable attitude toward that behavior [19]. For example, the belief that seeking treatment for a particular illness will lead to social ostracization is a common example of unfavorable attitudes toward health-seeking and a predictive deterrent of preventive health behaviors. In Kenya, perceived stigma has consistently been reported as a barrier to health-seeking and prevention for a variety of health issues [20,21,22].

In addition to influential constructs related to behavior, literature on health behaviors and decision-making among PLW point to several groups that influence the decision-making process of PLW. One such group prominent in the Kenyan literature are male spouses and partners, whose interdependence, communication, and participation with their pregnant or lactating partners have been found to affect behaviors such as diagnosis disclosure and adherence to medication [23,24]. Health providers are also a key population who have consistently been reported as one of the most trusted sources of health information that can subsequently help inform health behavior [25,26,27,28,29]. Moreover, pregnant women may be up to 12 times more likely to receive a vaccine when recommended by their health provider [30]. Although much of this research has been conducted in the context of HIV, influential parties are likely of similar importance in the context of COVID-19. Therefore, the aim of this study is to understand how three key audiences–pregnant and lactating women (PLW), male community members, and health workers–in Kenya conceptualize COVID-19 to better understand determinants of COVID-19 related behaviors. We aim to use the results of this study to generate communication strategies and inform public health programs and policies in Kenya to encourage recommended COVID-19 behaviors.

## 2. Materials and Methods

This study used qualitative methods to conduct 84 in-depth interviews with three sets of audiences: (1) 31 PLW, (2) 33 community members, including male family members and neighbors of PLW and community gatekeepers, and (3) 20 health providers, including nurses, midwives, community health volunteers, and other service providers. This research was conducted in a total of 6 communities in both urban and rural settings across three counties: Garissa, Kakamega, and Nairobi. We determined approximate sample sizes based on study aims and qualitative theory. This study used a grounded theory approach, in which sample size is determined based on data saturation rather than quantity of data [31,32,33]. Grounded theory saturation was reached when no new themes were emerging from the data. See Table 1 for participant information.

In-country study staff leveraged existing infrastructure to identify and recruit participants. Participants were identified in several ways, depending on audience type. PLW and community gatekeepers were identified in collaboration with community health workers, county health officials, and government health facility staff; male community members were identified through snowballing by recruiting family members and neighbors of PLW already recruited into the study, as well as through referral by health facility staff; and health workers were identified by local and county health managers. Data collectors recruited participants at health facilities across the three counties using a screening form to determine whether or not the potential participant met the criteria of being 18 years or older, identifying as one of the target audiences (PLW, relative or community member of someone who is pregnant, or health worker providing maternal health services), and being in their second or third trimester of pregnancy (if applicable). Data collectors obtained oral consent from all participants.

Semi-structured interview guides were designed in collaboration between study staff at Johns Hopkins University and Jhpiego and were pre-tested in the field. Interview questions varied by participant type; however, all instruments assessed respondents’ knowledge, attitudes, behaviors, and experiences related to COVID-19 (see Appendix A for the in-depth interview guides). Data collectors with Jhpiego Kenya were trained during a three-day workshop on project objectives, qualitative research techniques, human subjects research ethics, and COVID-19 precautions. Data were collected from August to September 2021. Due to safety considerations resulting from the COVID-19 pandemic, a combination of in-person and virtual interviews were conducted. Interviews were conducted in English or Swahili, depending on participant preference, and were audio recorded, transcribed, and translated to English (if necessary) by study staff fluent in English and Swahili. All data were stored on encrypted servers.

Data were analyzed by a team of seven using Atlas.ti software. Applying a grounded theory approach, the team generated, refined, and agreed upon a code list through three rounds of open coding. The team then coded approximately 25% of the transcripts, met to discuss emerging themes, then coded the remaining 75% of transcripts. Following coding, two researchers conducted interrater reliability with ~10% of the transcripts, returning reliability of >90%. All researchers then met to discuss themes and sub-themes.

This study was approved by the Institutional Review Board with Johns Hopkins Bloomberg School of Public Health and the Scientific and Ethics Review unit with Kenya Medical Research Institute.

## 3. Results

Results are organized into five key themes: myths, risk perception, economic implications, stigma, and self-efficacy. We organized these themes based on behavioral constructs that emerged through the data thought to influence decision-making for COVID-19 related behaviors.

### 3.1. Myths

PLW were generally aware of the existence of COVID-19 including symptoms and prevention measures. However, community members and health workers expressed that despite health education efforts in their communities, awareness remains inadequate. Across all participant groups, myths relating to the existence, spread, and severity of COVID-19 were pervasive; the most commonly reported myth being that COVID-19 does not exist. A family member in an urban area of Kakamega alludes to this: “[COVID-19] hasn’t affected people because it’s not there”. A pregnant woman in urban Kakamega expanded on how this disbelief in COVID-19 shapes her community’s views on mortality: “People have not accepted if there is COVID-19, they just say, ‘if it is your time to die, you will die.’ That is the mentality of many people…They just say it is normal death. if you say someone has suffered from COVID, they don’t believe you”.

For many, this belief stemmed from limited spread in their community, as this family member in urban Kakamega demonstrates: “My belief is that I can’t get it… Why I can’t get it is that ever since last year they have said that [COVID-19] is there… I have never seen any of my neighbors or someone from a neighboring county, I haven’t seen them complaining that there’s anyone who has it”. One health worker from Nairobi articulated the pervasiveness of this myth in their community: “From the view of the people, we have had very few people succumbing to COVID, so the community here is a bit skeptical about COVID. When you talk to them, the first thing that will come up is, ‘We have not seen someone suffering from COVID, so we cannot believe in something that we have not seen.’”

Others, however, not only believed in the non-existence of COVID-19, but that it is “another way the government is using us to get money”, as a family member in a rural community in Kakamega put it. Another myth was the belief that COVID-19 is only transmissible to certain populations, primarily those that are urban, wealthy, Western, or Christian. This myth was particularly prominent in Kakamega County. As this pregnant woman in a rural area describes: “[COVID-19] looked like it was a disease of the urban. My husband used to tell me, ‘you and your children are going to infect us with COVID,’ and I would tell him, ‘COVID is not for the poor because we take so many rough things. COVID is for the people who are taking soft things, not me.’ But finally, [COVID] came in so fast. That is when I knew it was for all”. One health worker in urban Kakamega affirmed the prominence of this myth and went on to describe their efforts to combat it: “Here, right now, we are trying to educate society on COVID and its effects, of which most people here have not yet embraced. They say it is an ailment of the Chinese”.

Other myths that were reported to a lesser extent included the perception that COVID-19 is not a cause for concern and can be treated with homeopathic remedies. For example, this pregnant woman from Garissa says: “If I was told I have COVID-19, I would basically take the herbal things that we usually have at home and steam myself and try to see if I am going to get well from that”. A health worker in Nairobi described in detail the natural remedies they have been taught to treat COVID-19 and how those remedies cured their child: “There is a lot of natural care. We have onions, ginger, garlic, and so forth. They really help so much. When somebody does it well, they don’t reach the ICU… I also massage the whole body with the rub. The whole body you stretch from the spine, the neck, the thighs. [The doctors] told us that COVID holds the thighs so tight, that is why it puts you down. So, we massage the legs… When my baby tested positive with symptoms, [the doctor] told us to put the onions on the armpit, on the socks and even on the stomach. We used everything. I put tomatoes all over and within two days we could not believe [the results]”.

### 3.2. Risk Perception

#### 3.2.1. Perceived Susceptibility

The perception that everyone is at risk of contracting COVID-19 because “it does not discriminate” (lactating mother, urban Kakamega) was a common finding across PLW and community members. As this family member in rural Kakamega poignantly states: “[I am at risk] because I am a human being and I breathe and I have a future…They say that it is an airborne disease and I breathe that same air. You can be any person, you will still get infected”.

Pregnant women and lactating mothers spoke to their unique susceptibility to the disease for various reasons. For one, pregnant women and lactating mothers perceived a risk of transmission from mother to fetus or child. As these lactating mothers put it: “If [COVID-19] infects a breastfeeding mother like me, the baby will definitely be affected” (Nairobi) and “If I get Corona, my child will also get it… And also the expectant mother. If she gets [Corona], even the baby inside will get Corona” (urban Kakamega). Additionally, pregnant women and lactating mothers both perceived themselves and were perceived by others as having greater susceptibility to COVID-19. This pregnant woman from rural Kakamega describes the risks of socializing for PLW: “[Pregnant women and breastfeeding mothers] are in danger because when they congregate somewhere and they do not observe social distancing, they can be infected”. A family member from urban Kakamega reinforced this notion: “The reason as to why I say that [pregnant women] are at a bigger risk of contracting Corona is because when they walk around and meet other people, you know women like talking and sitting together, laughing, and touching each other”.

Health workers perceived a high degree of susceptibility due to the nature of their work. Although most health workers “try to do self-care” by wearing proper PPE while working, “there are so many things that can make us get it” (health worker, Nairobi). Despite these findings, there remained several participants who perceived little risk of contracting COVID-19. For example, this family member from Nairobi shares: “I am not thinking that I will get Corona. First of all, the protocols and guidelines that have been put in place by the health workers is that each one of us should take personal responsibility. You should always wear a mask, keep on washing your hands. So if you take care of yourself, your chances of getting Corona are minimal”.

#### 3.2.2. Perceived Severity

Perceptions of severity were fairly consistent across target audiences. To nearly all who believed in COVID-19, it was seen as a “deadly” disease. As one lactating mother from urban Kakamega put it: “COVID is deadly. You can suffer for 3 days or even just a day and then you succumb to it… It causes death and it leaves other people behind who are so helpless”. Health workers echoed this notion, as one from Garissa shared: “[COVID-19] can kill you instantly if you joke with it. So, I think it is very severe. It is a killer disease”. For this gatekeeper in Nairobi, the lethality of COVID-19 is readily apparent: “It is now said that COVID-19 has taken center stage because we no longer have a hearse which is not booked”.

Some participants elaborated on the severity of the disease, noting how severity can be exacerbated by co-occurring conditions. For instance, this family member from Nairobi shared this belief: “People say that if you have other underlying illnesses, it will kill you”. A neighbor from rural Kakamega echoed this fear: “You know, we have different diseases. You may get a heart attack or cancer, so if you are infected and you have an underlying condition, you may not make it. That is the fear we have at the moment”.

### 3.3. Economic Implications

Aside from the health implications of the COVID-19 pandemic, participants described several non-health related challenges that have emerged. Economic downturn and financial hardship resulting from the pandemic was a recurring theme across all participants. As one family member from Nairobi summarized: “Before Corona came, business was good, life was good, even traveling was good. Everything was OK. But when it came, it destroyed everything. The economy has crumbled, people have lost their lives, they have left their families, and you cannot interact with your family well”.

When asked how COVID-19 has affected their community, participants most commonly reflected on employment. This lactating mother from urban Kakamega describes her personal experience with economic hardship resulting from the pandemic: “The impact of COVID-19 on my daily responsibilities as a young woman, I have to close down my business… If I am a working-class lady, I will have to at least miss going to work. I will be given compulsory leave and when you come back you find that you have already been replaced at the job. You find you are back to stage one… So with COVID, you will not even be able to protect your job, to protect your daily income, to protect the people you love, and also your daily life. It will all be affected”. Similar stories of job loss and business closures were widespread across the data.

Although many participants attributed these challenges to self-isolation from a positive test result, others pointed to certain aspects of the community-level pandemic response. In particular, several participants suggested the nation-wide curfew, negatively impacting local businesses. This lactating mother from rural Kakamega describes her experience: “COVID has negatively impacted our economy. Recently, there has been a curfew and many people survive from hand-to-mouth. They need to go out to work and you realize that people are told to stay at home. There are those people who work at night, but they are told that the curfew begins at 7:00 P.M. There are those who sell vegetables who come to the market at 6:00 P.M. and they expect to be selling up until 8:00 P.M., yet curfew begins at 7:00 P.M. That has really impacted our economy”.

In addition, pregnant women, lactating mothers, and health providers reported experiencing or observing others struggle with medical care costs. As this lactating mother from Nairobi put it: “Yes, [COVID-19] is a problem because if you contract it, there are expenses. The expenses of taking care of a COVID-positive person is so expensive”. A health worker in Garissa elaborated on this dilemma, explaining that: “People are spending a lot of money. Some have traveled to hospitals in Nairobi, some are spending many days in the wards in our referral center, and some are dying. So, it is impacting the community in many ways”.

### 3.4. Stigma

Participants described a societal inclination to avoid and, in some cases, completely ostracize individuals who contract COVID-19 to avoid transmission. Societal stigma against COVID-positive people was a prominent theme that appeared across all target audiences and geographies. As this family member from Nairobi summarized: “[A COVID-positive person] is viewed as someone who is no more. You are isolated. There is no one who will come near you”. This lactating mother from Garissa shared the extent to which her community fears even the objects touched by a COVID-positive person: “[The community] cannot accept him. Even now as we speak, they are scared of the cup he used to drink from… just like the disease itself”.

In many cases, this stigma permeated families to the point that, even following recovery, spouses and other family members were afraid to re-integrate them into the home. As this health worker from rural Kakamega described: “[Families] normally abandon [relatives with COVID-19]. They won’t want to associate with them. There was one case when the patient was discharged, the relatives told us they don’t want that client to go back home because she will infect them. So, we had to arrange for a facility-based care at another facility until the patient recovered”.

For several participants, the tendency to stigmatize COVID patients was compared to former stigmatization of HIV-positive people in their communities. As this lactating mother from Garissa put it: “In the past, there was stigmatization of HIV, but nowadays, it is Corona. If you hear that someone in your family has died or recovered from Corona, people will not even come to greet you”. One gatekeeper from Nairobi expanded on this, explaining how comparing COVID-19 to HIV can have a stigmatizing effect on the entire family: “Initially there were a lot of myths… Just as somebody with HIV was taken as a burden by the community, so is the person with COVID-19, because the moment COVID-19 is prone in a family or household, that would be the start of the end to the family. In terms of the care and support that these people require financially, physically, emotionally, spiritually, it is not easy. It would be seen as a draining of resources to this particular family… So people are like, ‘Can we leave them for dead?’ and so on”.

Stigmatization of COVID-positive people as a form of prevention was noted as having harmful, lasting implications for those stigmatized. Participants expressed that the marked manners in which their communities are responding to COVID patients and survivors is causing degradation of their mental health to such an extent that, “when we isolate [COVID-positive people], we kill them” (health worker, Nairobi). This family member from urban Kakamega shared how he imagines stigma from COVID-19 evolves: “One feels that there is no hope to live. Their self-esteem goes low. Some will feel like they don’t have the immune system to fight Corona, so he pulls himself down and maybe it was not time for him to die, but depending on how that person’s mentality is, it may kill him because he will feel he does not have hope and he panics”.

### 3.5. Self-Efficacy

In terms of one’s perceived control over protecting oneself and others from contracting SARS-CoV-2, participants had differing perspectives. In some cases, PLW and male community members described feelings of powerlessness and inevitability which stemmed from religious beliefs. As one male family member succinctly put it: “[COVID-19] is something that only God can prevent… COVID-19 here in Garissa is very high, so we just pray to God for help”. A lactating mother in Nairobi agreed, stating: “[COVID-19] is a problem because when it’s in the community, it’s only God who helps. Others will get well while others will die”. In other cases, participants reported structural barriers which prevented them from adhering to risk-reducing activities. For some, the costs of necessary products like soap and sanitizer were unaffordable: “You will find that somebody can easily lose his or her life just because of Ksh.10” (gatekeeper, rural Kakamega). For others, the task of maintaining social distancing was unrealistic given their community’s living conditions. As one community member in Nairobi stated: “In these one-roomed houses with children and parents, it is a very big challenge. You maybe share toilets, bathrooms, shops. Basically, you do things together, so it is very easy for the disease to spread in the community”.

Health workers also described a lack of control when it came to preventing transmission in their health facilities. Despite providing protective and hygienic resources such as masks, soap, and hand sanitizer to patients and other visitors, health workers reported low use of these resources among the general public. One health worker from Garissa described their frustrations with trying to curb transmission in their facility: “Even at the hospital, those who come do not put on masks. So, it is left up to us (health workers) to tell them to put on masks… We have washing stations for hand wash, and they do not wash their hands. We have sanitizers and yet, we are the ones to tell them what to do”.

Despite these instances of lower self-efficacy, many participants described how they take control of their risk by adapting their behaviors. PLW often reported that they “don’t go to places where there are sick persons” (lactating mother, Garissa) and “always have masks on especially when you are in a public place” (lactating mother, Kakamega). One family member in Nairobi explained how taking ownership of one’s own preventive behaviors can encourage others to act similarly: “First of all, you are the one who is supposed to be safe… You have to protect yourself before you tell the others. So, if you are wearing a mask, that is when you can tell your neighbor that Corona is real, that they should wear a mask, sanitize, keep their home environment clean. That is when they will heed”.

## 4. Discussion

This study presents the various ways in which COVID-19 is conceptualized among PLW, male community members, and healthcare workers in three Kenyan counties through myths, perceived risk, financial implications of the pandemic, stigma, and self-efficacy.

Myths related to COVID-19 were prominent among pregnant and lactating participants and the communities in which they reside. This finding supports prior research which has identified common misconceptions relating to the existence, spread, and weaponized use of COVID-19 in Kenya and the Sub-Saharan African region, suggesting misinformation and misconceptions about COVID-19 are not uncommon in Kenya [34,35]. Researchers have established several ways to mitigate harm-causing rumors, such as those experienced in Kenya. One strategy that has shown promising preliminary results is the use of e-health, including chatbots and mobile apps, to answer questions related to the COVID-19 vaccine [36,37,38,39,40,41]. This e-health approach offers a potential means of disseminating accurate health information virtually during a time when in-person interactions pose a substantial risk. However, in Kenya, e-health initiatives for the COVID-19 vaccine must address previously identified limitations with e-health in the country. For example, Njoroge et al. point to the needs for e-health initiatives in Kenya to expand their access to rural and other marginalized areas and ensure a commitment to evaluating program effectiveness [42]. Moreover, the use of chatbots to combat COVID-19 is still a highly novel technique and challenges with user acceptance, fact-checking, and reaching communities without internet are evident [36]. Other potential strategies to mitigate myths related to the COVID-19 vaccine include community education campaigns to improve vaccine literacy and increase awareness of myths, identifying and disseminating health information to communities’ most-trusted sources of information, developing partnerships between public health entities and social media or news outlets, and leveraging the voices of social leaders [34,35,43,44,45].

Participants perceived PLW as well as healthcare workers to be highly susceptible to SARS-CoV-2 infection. Similarly, studies across the Sub-Saharan Africa region have shown high risk perception of COVID-19 among healthcare workers [46,47,48]. In one study, Girma et al. found that health professionals working in public university hospitals in Ethiopia reported greater perceived vulnerability to COVID-19 than other prevalent infectious diseases in the region, including HIV, malaria, and tuberculosis, underscoring these findings [47]. Few studies have explored risk perception among PLW in this region; however, research across other areas of the world indicate that fear of infection and harm to the pregnancy are common concerns [12,49,50]. Such studies have found that pregnant women with high risk perception may also display protective behaviors such as vigilance for transmission, modifying workplace responsibilities, or information-seeking and may also experience symptoms of stress, anxiety, and depression [12,49,50]. Participants also displayed awareness of the severe manifestations of COVID-19 and the extensive impact the pandemic has made on their communities and across the globe. Of note, participants spoke of the deadliness of COVID-19, often describing personal experiences with family loss resulting from the pandemic. These findings may have implications for the public’s adherence to preventive strategies (i.e., mask-wearing, social distancing, hand washing, etc.), as several studies point to the positive association between risk perception and compliance [13,50].

Program implementers and health communicators should consider leveraging risk perceptions among target audiences in this study to promote compliance with behavioral prevention. Several strategies could be used to do so, such as providing factual information about the burden of COVID-19, using empathy to connect with and express concern for an audience, and communicating a direct and clear connection between behavioral prevention and reduced risk of disease [51,52]. However, Li et al. make an important declaration in their article on how risk perception motivates preventive behavior during the pandemic in the U.S. and China, which is that the relationship between risk perception and behavioral prevention is not always unilateral and can be complicated by contributing factors such as social and cultural norms [53]. It is essential that implementers targeting risk perception move quickly and adaptably to meet the unique needs of their target audience.

Economic hardship resulting from both individual experiences with COVID-19 and community-wide preventive measures was widely experienced across this sample. Namely, participants spoke about the impact of the pandemic on job loss and restrictions to local businesses, causing families to struggle to make ends meet. According to participants, job insecurity was a result of both contracting the disease and the recovery process as well as local prevention measures which restricted business hours and consumer travel. The pandemic’s impact on the economy has not only been felt in Kenya, but across the region [54,55,56,57]. Experts point to Ghana as an example of how countries like Kenya might improve and maintain their economic stability [57]. In the early stages of the pandemic, Ghana displayed flexibility in its policies by opening up domestic trade and offering transparency to its citizens on the country’s financial status. Rather than imposing strict regulations to control the population, Ghana focused on community education and implemented creative solutions to funding prevention campaigns [57]. Until the concerns brought up by participants regarding the job market, local business restrictions, and costs of medical care are addressed in Kenya, communities, including PLW, will continue to face significant hardship. Officials in Kenya might consider implementing an approach which strikes a balance between offering greater flexibility with local businesses to improve the economy and reducing barriers to behavioral prevention by increasing access to and reducing costs of masks, sanitizer, and tests. In their multi-national review of countries’ economic interventions to address the COVID-19 pandemic, Danielli et al. identified several ways in which countries have adapted to meet the financial and health needs of their citizens [58]. Strategies ranged in flexibility and scale and included subsidizing a specified percentage of employees’ salaries to protect businesses while creating room for individuals to afford necessities, suspending mortgage payments, and providing loans and grants to business for protection. Although their findings were specific to high- and middle-income countries, the lessons learned may be applicable to Kenya.

Participants also described pronounced stigma against those who test positive for COVID-19, often compared to the ostracization once experienced by HIV-positive people in their communities. According to our data, Kenyan COVID-19 survivors may experience detrimental effects on their mental health and their social support networks as a result of stigma within their families and broader communities. This stigma has the potential to discourage preventive behaviors critical for controlling the spread of the pandemic, such as screening, social distancing, using proper hygiene, and seeking treatment for positive cases [59,60]. Moreover, recent studies point to stigma as a barrier to disclosing one’s COVID-19 status to those around them, putting others at risk of contracting the disease [48,61,62,63]. Several recommendations have been posited to address the stigma related to COVID-19. As many suggest, tackling stigma requires a multi-level approach, utilizing strategies at the policy level (such as dedicating public offices to reducing stigma and discrimination), community level (such as monitoring mass media platforms and mobilizing community leaders), and individual level (such as seeking self-education and engaging in conversations with empathy) [64,65,66]. Considering the parallels participants described between stigma toward COVID-19 and HIV in Kenya, researchers and program implementers might consider adapting an approach informed by successes from previous HIV work in the country. For instance, such studies often targeted patients’ interpersonal relationships with family members, members of their social network, and health providers to focus on holistic wellness and social support [64,65,66].

Self-efficacy varied among participants, with some describing an inability to adhere to behavioral prevention due to religious beliefs and structural barriers, while others described instances in which they have taken control of their risk of transmission by complying with recommendations. Little research has been done in Kenya to identify structural barriers to nonpharmaceutical prevention strategies. However, one study (preprint) of more than 2000 participants in informal settlements in Nairobi found that despite high awareness of their risk of transmission (83%), 37% and 53% of participants reported access to water and costs as barriers to hand washing and sanitizing, respectively [67]. Across the region, perceived control of transmission has been associated with greater uptake of behavioral prevention [68,69,70,71]. Considering the lack of research in this area in Kenya, identifying and understanding factors which can facilitate self-efficacy will be instrumental to improving behavioral prevention. These findings point to a need to address both perceived and actual control over behavioral prevention among PLW and other target audiences in Kenya. As the national government revises or contributes to their practical recommendations throughout the pandemic, they should consider the feasibility of such recommendations and the resulting impact on citizens’ self-efficacy.

This study is not without limitations. This cross-sectional qualitative study was not designed to be generalizable, and so findings cannot be attributed to settings outside the three counties studied in Kenya and conclusions related to residence (urban or rural) cannot be made. Given the self-reported nature of the study design and the somewhat divisive topic of COVID-19, social desirability bias is likely. Due to the recruitment strategy in which participants were identified through health and family networks, sampling bias is possible. The study was collected during a particular point in time (August–September 2021) of the SARS-CoV-2 pandemic, when knowledge about the disease and governmental vaccination policies were still evolving. During data collection, for example, the Kenyan Ministry of Health issued a press statement that stated that PLW could choose to receive COVID vaccines after counseling, which differed substantially from the previous recommendation which stated that PLW were not eligible for COVID vaccination (February 2021). Additionally, data were collected when COVID-19 cases and deaths were starting to decline in Kenya.

Despite these limitations, this study has several strengths. This is one of the few studies that has examined COVID-19 related behaviors specific to PLW and provides additional insight about community members and health workers. Moreover, this study contributes to a growing body of literature exploring social, emotional, and behavioral experiences and outcomes during the COVID-19 pandemic. Findings from this study can and should be used to inform the design and development of tailored communication programs and policies relating to PLW, healthcare workers, and the general public in Kenya.

## 5. Conclusions

Pregnant women are at high risk of adverse outcomes following SARS-CoV-2 infection, and uptake of preventive behaviors are essential to their and their babies’ health and well-being during the pandemic. To facilitate health prevention among this group, we must first understand how PLW and key stakeholders in their decision-making processes experience and conceptualize COVID-19. This study sought to understand how three key audiences–PLW, male community members, and health workers–in Kenya have come to conceptualize COVID-19 to better understand determinants of COVID-19 related behaviors. Results indicate that risk perception and behavioral attitudes are largely influential in their experiences of the pandemic. The spread of misinformation is contributing to the continued presence of myths which are lessening the public’s perceptions of risk to COVID-19 and adherence to preventive behaviors. Moreover, fear of stigma and economic instability were commonly reported as barriers to testing, treatment-seeking, and social distancing. Addressing potential barriers to preventive health behaviors, such as the spread of misinformation, financial constraints, and fear of social ostracization, will be essential for future public health prevention and communication responses targeting these groups.

## Figures and Tables

**Table 1 ijerph-19-10784-t001:** Participants by audience type and location.

	Garissa (Rural)	Kakamega (Rural and Urban)	Nairobi (Urban)	Total
Pregnant and lactating women	8	10	11	29
Community members (male family members, male neighbors, community gatekeepers)	8	12	15	35
Health workers (nurses, midwives, community health volunteers)	6	8	6	20
Total	22	30	32	84

## Data Availability

Data presented in this study are available in this article. Further data sharing is not available.

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
