# Peer review of "Conceptualizing the COVID-19 Pandemic: Perspectives of Pregnant and Lactating Women, Male Community Members, and Health Workers in Kenya"

_ijerph, 2022, doi:10.3390/ijerph191710784_

Round 1

Reviewer 1 Report

In this article the authors propose to analyze how three key audiences– pregnant and lactating women (PLW), male community members, and health workers– in Kenya conceptualize COVID-19 to better understand determinants of COVID-19 related behaviors. This study used qualitative methods to conduct 84 in-depth interviews in three counties in Kenya.

The Authors deal with a particularly interesting and still not much debated topic in literature.

English is quite good.

The abstract summarizes the purpose well, and reports the results obtained.

The introduction is well written and addresses COVID-19-related issues, especially in reference to the Kenyan population.

In my opinion, the parts to improve are the materials and methods: 84 interviews seem few to me; furthermore, it is not clear how the interviewees were selected; Were any tests done before administering the questionnaires ?; statistics processing in my opinion is lacking and should be improved; in addition, questionnaires should be included as supplementary material.

In the discussion, the authors should better specify the practical implications of this study and the limitations should be broadened as recommended.

In my opinion, the article requires a revision before being considered for publication on IJERPH.

Author Response

Dear reviewer, 

Thank you for your insightful comments on our manuscript “Conceptualizing the COVID-19 Pandemic: Perspectives of pregnant and lactating women, male community members, and health workers in Kenya." We appreciate your review and believe we have addressed all of your concerns. Please find the responses to your comments below and a revised manuscript with tracked changes attached.

Reviewer 1

Point 1: In my opinion, the parts to improve are the materials and methods: 84 interviews seem few to me; furthermore, it is not clear how the interviewees were selected; Were any tests done before administering the questionnaires ?; statistics processing in my opinion is lacking and should be improved; in addition, questionnaires should be included as supplementary material.

Response 1:

  • As this was a qualitative study, it does not have the same requirements related to sample size calculations. Generally, the sample for qualitative studies is based on what is being studied and data saturation, and 84 interviews is considered a large sample for a qualitative study. We used a grounded theory approach which suggests 20-30 participants. (See Morse, J. M. (2000). Determining sample size. Qualitative health research10(1), 3-5.).
  • Regarding statistics processing, this was not a quantitative study. We conducted inter-rater reliability to assess coding agreement – this information is included in the manuscript.
  • Regarding how interviewees were selected and if there were any tests done before administering questionnaires: We have added the following statement to the methods section to clarify the process by which participants were identified and screened:

“In-country study staff with Jhpiego Kenya leveraged existing infrastructure to identify and recruit participants. Participants were identified in several ways, depending on audience type. PLW and community gatekeepers were identified in collaboration with community health workers, county health officials, and government health facility staff; male community members were identified through snowballing by recruiting family members and neighbors of PLW already recruited into the study, as well as through referral by health facility staff; and health workers were identified by local and county health managers. Data collectors recruited participants at health facilities across the three counties using a screening form to determine whether or not the potential participant met the criteria of being 18 years or older, identifying as one of the target audiences (PLW, relative or community member of someone who is pregnant, or health worker providing maternal health services), and being in their second or third trimester of pregnancy (if applicable). Data collectors obtained oral consent from all who were interested in joining the study.”

  • Regarding the questionnaires: The two qualitative interview guides used for this study have been uploaded as supplementary materials. One interview guide is for community members (pregnant women, lactating women, other community members) and one is for health providers.

Point 2: In the discussion, the authors should better specify the practical implications of this study and the limitations should be broadened as recommended.

Response 2:

  • Regarding the practical implications of the study: We have revised the discussion section of the text to elaborate on the practical implications of the study. While we felt we had adequately addressed the practical implications relating to myths and stigma, we agreed that more detail was needed relating to risk perception and economic hardship. In the added text, we reference relevant literature and provide recommendations for how to address these issues in practice. We have also included new text to discuss the implications of the self-efficacy theme and include practical recommendations.
  • Regarding the limitations: We have revised the limitations section of the text to further explain some existing limitations as well as add new limitations regarding sampling and analysis. We hope this more accurately captures the study’s constraints.

Reviewer 2 Report

Dear Authors

With great interest I read your study about determinants of COVID-19 related behaviours of three different audiences in Kenya. Results indicate that risk perception and behavioural attitudes are largely influential in their experiences of the pandemic. The spread of misinformation is contributing to the continued presence of myths which are lessening the public’s perceptions of risk to COVID-19 and adherence to preventive behaviours.

In my opinion, the manuscript is well written, and you address an important topic with great impact to almost every country worldwide: How to stop the COVID-19 pandemic and understand its preventive measures, diagnosis, and therapies.

I have only minor concerns and would then recommend the manuscript for publication.

Introduction:

Kenya is a religious country: What`s the authors opinion about the influence of religion in terms of behavioural attitudes and spread of (mis)-information about COVID-19?

Line 49: missing space after “populations”

Methods

Can you provide the questionnaires as supplementary material?

Results:

Line 412: COVID-19

What were these three groups of people thinking about wearing face masks? Were facemasks widely available during the time of interviews?

Was there a difference in risk perception and behavioural attitudes between the countryside and bigger cities?

Author Response

Dear reviewer, 

Thank you for your insightful comments on our manuscript “Conceptualizing the COVID-19 Pandemic: Perspectives of pregnant and lactating women, male community members, and health workers in Kenya." We appreciate your review and believe we have addressed all of your concerns. Please find the responses to your comments below and a revised manuscript with tracked changes attached.

Reviewer 2

Introduction

Point 1: Kenya is a religious country: What`s the authors opinion about the influence of religion in terms of behavioural attitudes and spread of (mis)-information about COVID-19?

Response 1: This is an excellent question and one that we asked, as well. From the data, we found few instances in which participants’ views of the COVID-19 pandemic were influenced by religion. Namely, these participants described a sense of powerlessness, such that only God can control who will get sick and how the pandemic will turn out. These comments were very few, which is why we did not identify religion as a prominent theme or touch upon it in the original manuscript. However, after reconsidering this question (as well as your question below about adherence to masks), we believe this topic can be addressed through a new theme which was prominent: self-efficacy. We have added this theme to our results and discussion sections and hope it fills the gap in our paper.

Please note that participants did not report any instances of religion directly influencing their attitudes toward nonpharmaceutical behavioral interventions or being a source of (mis)information related to COIVD-19. However, other data from this study did indicate a substantial impact of religion on COVID-19 vaccine decision-making. These results are reported in another manuscript currently under review.

Point 2: Line 49: missing space after “populations”

Response 2: Thank you for noticing this. A space has been added.

Methods

Point 3: Can you provide the questionnaires as supplementary material?

Response 3: The two qualitative interview guides used for this study have been uploaded as supplementary materials. One interview guide is for community members (pregnant women, lactating women, other community members) and one is for health providers.

Results

Point 4: Line 412: COVID-19

Response 4: Thank you for noticing that the 19 was missing. We have corrected this.

Point 5: What were these three groups of people thinking about wearing face masks? Were facemasks widely available during the time of interviews?

Response 5: According to reports from health workers, masks, soap, and sanitizer were being widely distributed at health facilities. However, reports on adherence to mask-wearing and other forms of behavioral prevention were mixed. Despite availability of needed products at health facilities, participants reported structural barriers to practicing behavioral prevention, such as cost and inability to social distance. We have expanded upon this in the results section under the new theme: self-efficacy.

Point 6: Was there a difference in risk perception and behavioural attitudes between the countryside and bigger cities?

Response 6: Due to the study’s sample size, we were unable to make direct comparisons across geographical areas. Although we observed few differences across urban and rural populations in our analyses, we believe a larger sample size is needed to confidently report these conclusions. This has been added as a limitation in the discussion section. We did attempt to assess differences in rural versus urban participants, but there were no major differences in the emerging themes presented.

Round 2

Reviewer 1 Report

The authors have greatly improved the article. In my opinion, the question remains about the limited number of participants in the questionnaire. On this aspect I still have some doubts. In my opinion the Authors should emphasize this point in the limitations.

Author Response

Dear Reviewer 1, 

Thank you for your kind words and additional feedback. We have provided our response below and revisions in the tracked document. 

Reviewer 1

Point 1:  The authors have greatly improved the article. In my opinion, the question remains about the limited number of participants in the questionnaire. On this aspect I still have some doubts. In my opinion the Authors should emphasize this point in the limitations.

Response 1: Although we appreciate this reviewer’s concern for sample size, we disagree that this is a limitation of the research. Unlike in quantitative research, qualitative research (especially that which follows grounded theory) does not aim for data repetition through large sample sizes [1-3]. Instead, it aims to collect rich, high-quality data which illustrate recurring themes until the point at which no new themes emerge, otherwise known as “data saturation” [2,3]. In fact, it is recommended that grounded theory studies have a sample size which ranges from 20-30 interviews [4,5]. In this study, data saturation was reached within 84 interviews. Therefore, additional interviews were not required to identify themes and draw conclusions.

We have clarified this decision-making process in the methods section and added appropriate references.

  1. Vasileiou, K.; Barnett, J.; Thorpe, S. et al. Characterising and Justifying Sample Size Sufficiency in Interview-Based Studies: Systematic Analysis of Qualitative Health Research Over a 15-year Period. BMC Med Res Methodol 2018, 18, 148, doi:10.1186/s12874-018-0594-7.
  2. Charmaz, K. Constructing Grounded Theory: A Practical Guide Through Qualitative Analysis; Sage: London, England, 2006.
  3. Bowen, G.A. Naturalistic Inquiry and the Saturation Concept: A Research Note. Qual Res. 2008; 8, 137–52, doi:10.1177/1468794107085301.
  4. Marshall B, Cardon P, Poddar A, Fontenot R. Does sample size matter in qualitative research?: a review of qualitative interviews in IS research. J Comput Inform Syst. 2013;54(1):11–22.
  5. Morse, J. M. (2000). Determining sample size. Qualitative health research10(1), 3-5.